

# Assessment of warm-water coral reef tipping point thresholds

Paul Pearce-Kelly[1], Andrew H. Altier[2], John F. Bruno[3], Christopher E. Cornwall[4], Melanie McField[5], Aarón Israel Muñiz-Castillo[5], Juan Rocha[6], Renee O. Setter[7], Charles Sheppard[8] Rosa Maria Roman-Cuesta[9], Chris Yesson[10]

[1] Zoological Society of London, Regent's Park, London NW1 4RY, UK paul.pearce-kelly@zsl.org
[2] Department of Environmental Engineering Sciences, Engineering School of Sustainable Infrastructure & Environment, University of Florida, USA. andrew.altieri@essie.ufl.edu
[3] Department of Biology, The University of North Carolina at Chapel Hill, Chapel Hill, NC, USA jbruno@unc.edu
[4] School of Biological Sciences and Coastal People Southern Skies Centre of Research Excellence, Victoria University of Wellington, Kelburn, Wellington, New Zealand. christopher.cornwall@vuw.ac.nz
[5] Healthy Reefs for Healthy People Initiative, Mexico, Belize, Guatemala, Honduras, and the USA. Fort Lauderdale, FL 33312, USA.mmcfield@gmail.com israbios.muca@gmail.com
[6] Stockholm Resilience Centre, Stockholm University, Stockholm, Sweden, juan.rocha@su.se
[7] Department of Geography and Environment, University of Hawai'i at Mānoa, Honolulu, Hawai'i, USA. rsetter@hawaii.edu
[8] University of Warwick, University Road, Coventry, CV4 7AL, UK. Charles.Sheppard@warwick.ac.uk
[9] Technical University of Munich, School of Life Sciences, Freising, Germany. rosa.roman@tum.de
[10] Institute of Zoology, Zoological Society of London, Regent's Park, London NW1 4RY, UK chris.yesson@ioz.ac.uk

*Correspondence to*: Paul Pearce-Kelly (paul.pearce-kelly@zsl.org)

**Abstract**. Warm-water coral reefs are facing unprecedented Anthropogenic driven threats to their continued existence as biodiverse, functional ecosystems upon which hundreds of millions of people rely. Determining the tipping point thresholds of coral reef ecosystems requires robust assessment of multiple stressors and their interactive effects. We draw upon a literature search and the recent Global Tipping Points Revision initiative to consider warm-water coral reef ecosystem tipping point threshold sensitivity. Considering observed and projected stressor impacts we recognise a global mean surface temperature (relative to pre-industrial) tipping point threshold of 1.2°C (range 0.7-1.5°C) and an atmospheric $CO_2$ warming threshold of 350ppm (range 326-400 ppm), whilst acknowledging that interacting stressors, ocean warming response time, overshoot and cascading impacts have yet to be sufficiently assessed but are likely to lower this threshold. These uncertainties around tipping point sensitivities for such a crucially important ecosystem underlines the imperative of robust assessment and, in the case of knowledge gaps, employing a precautionary principle favouring the lower range tipping point values.

## 1. Introduction

Warm-water coral reefs (comprising tropical and sub-tropical reefs) are estimated to support a quarter to one third of marine biodiversity (Plaisance et al 2011), including over 25% of marine fish species, and annually provide nearly US$9.8 trillion worth of ecosystem services (IUCN 2016), upon which at least 500 million people are reliant (IPBS 2019). They are also among the most sensitive ecosystems to anthropogenic driven stressors with an estimated 50% of global live coral cover having been lost over the last 50 years (Souter et al 2021, WWF 2022), primarily due to ocean warming (and related climate change



threats of ocean acidification and deoxygentation), but in some locations also due to fishing, pollution, and disease (IPCC
2022). IPBES (2019) states that over 80% of the world's coral reefs are severely over-fished or have degraded habitats
(McClanahan et al., 2015). Eddy et al (2021) estimate the capacity of tropical and sub-tropical reefs to provide ecosystem
services has declined by half since the 1950s. Although local stressors continue to have profound impacts on coral reef health,
climate change driven stressors have become the dominant threat to the functional viability of these ecosystems and the
essential services they provide to hundreds of millions of people (IPBES 2019, IPCC 2022).
It is well established that coral reef ecosystems are vulnerable to multiple interacting tipping points (Norstrom et al 2016;
Heinze et al 2021, Armstrong-McKay et al 2022; IPCC 2022). IPCC (2022) defines a tipping point as *a critical threshold*
*beyond which a system reorganises, often abruptly and/or irreversibly*. Coral reefs are prone to tipping points that can
produce coral die offs (e.g. bleaching) and subsequent replacement by other ecological communities such as macroalgae,
soft corals, urchin barrens or corallimorpharians (Notrström 2016; Holbrook et al., 201), with low resilience, reductions in
biodiversity and degradation of ecosystem services (IPBES 2019). Warm water coral reefs cross a threshold of ecosystem
collapse (Bland et al. 2018) when they cease to have sufficient live coral cover (typically ~ 10%) necessary for supporting the
wide diversity of taxa, ecological interactions and positive carbonate production state typical of a coral reef (Darling et al 2019;
Perry et al 2013; Sheppard et al 2020; Vercelloni et al 2020; Armstrong-McKay et al 2022).  Mortality of corals may play out
over weeks to a few months for acute events (e.g. thermal stress-induced bleaching), or years for chronic threats (e.g. diseases
and land-based impacts), but prolonged failure to recover over a decade or more is necessary to qualify a coral reef as
'collapsed'. Coral reef collapse is an ecological phenomenon at local scales; here we explore where localised coral reef collapse
aggregates, potentially irreversibly, to regional and global scales.
Approximately half the live coral cover on coral reefs has been lost since the 1870s, with accelerating losses in recent
decades due to climate change exacerbating other drivers (IPBS 2019), with estimated loss of 16% in 1998 (Wilkinson et
al. 1999), measured loss of 14% from 2009 - 2018 (Souter et al 2020), and high variance among regions. Localised responses
of corals to increasing scales and intensities of stressors are aggregating at scales now exceeding 1000 km and manifesting as
regional die-offs (e.g. Western and Central Indian Ocean, Great Barrier Reef, Mesoamerican Reefs) (Le Nohaïc et al 2017;
Amir 2022; Muñiz-Castillo et al. 2021; Obura et al. 2022; Sheppard et al (2020), with most reef regions having experienced
multiple die-off events (Darling et al. 2019; Cramer et al. 2020; IPCC 2022). Coral reef bleaching tipping points have already
been reached in seven ocean systems (IPCC 2022).

## 2. Determinants for assessing coral reef tipping point thresholds

Direct and indirect local human activities are increasingly degrading coral reef ecosystems through a combination of coastal
development, water quality reduction by pollutant runoff and sedimentation, over-harvesting (especially fisheries), invasive



species and disease spread. At the local level, these stressors have proven sufficient to tip reefs into regime shifts from a coral
dominated ecosystem to a macroalgae dominated ecosystem (Bruno et al 2009; IPBES 2019; Souter et al 2021; Biggs et al
2018)e). Local stressor impacts are increasingly being eclipsed by anthropogenic climate change and can act synergistically
with climate change, for example, high abundance of macroalgae or urchins magnifying coral loss after bleaching (Donovan
et al 2021).

Interactions between different stressors can be antagonistic (the combined effect is less than the additive), additive (the
combined effect is equal to the sum of their individual effects) or synergistic (the combined effects exceed their individual
effects) (Good and Gahr 2020). Stressor onset rate can have a major effect on stressor impact as has been reported for coral
reef fish mortality (Genin et al 2020). Depending on their onset rate and magnitude, the same interacting stressors may initially
have antagonistic effects but may transition to having additive or even synergistic effects (e.g., Fisher et al 2019).

Increasing atmospheric greenhouse gas (GHG) concentrations, especially carbon dioxide ($CO_2$), are disrupting Earth Energy
Balance. The resultant Earth Energy Imbalance (EEI) is increasing atmospheric and ocean temperatures (IPCC 2021; Loeb et
al 202; Von Schuckmann et al 2020). $CO_2$ concentrations are the dominant driver of rate and magnitude of ocean warming and
acidification (Meinshausen et al 2020) with cascading effects on other coral reef stressors, most significantly marine
heatwaves, storm intensity, sea level rise, ocean deoxygenation and extreme climate events.

Ocean warming and ice-sheet melt respond slowly to any given level of $CO_2$ emissions and temperature with resultant
additional *committed* heating, sea level rise and resultant stressor impacts such as storm severity. Ocean warming response
time is approximately 20-30 years for the majority of committed warming to be realised (R. Betts personal communication 12
August 2023; IPCC 2021) and sea level rise commitment is over centennial time IPCC 2021). Due to this lag, tipping point
thresholds can be exceeded decades before the physical impacts are observed.

Overshoot describes warming pathways that temporarily increase global mean temperature over a specific temperature target
(IPCC 2022). Overshoot of multidecadal time spans imply severe risks and irreversible impacts in many ecosystems, including
coral reefs from heat-related mortality and associated ecosystem transitions (high confidence) (IPCC 2022).
Tipping point cascades describe a tipping point in one system triggering, or stabilising, subsequent tipping points in other
systems (IPCC 2022; Armstrong-McKay et al 2022; Rocha et al 2018; Wunderling et al 2023).
Here we summarise the most important factors in coral reef decline, summarising the major tipping points and interactions
between them.



## 3. Ocean warming and heatwaves

The primary driver of regional to global scale coral mortality and loss is marine heat waves (MHWs), which are caused by the interplay of the anthropogenic warming trend and natural variability of ocean temperature (e.g., the ENSO cycle that causes El Niño events). During tropical MHWs, ocean temperatures only 1–2 °C higher than the summer maxima to which corals are acclimatised can cause severe physiological stress leading to mortality via "coral bleaching". Although corals sometimes appear to recover from bleaching, growth rates and reproduction can be greatly reduced for years. Additionally, ocean warming is linked with some devastating coral diseases and appears to be increasing the frequency and intensity of cyclonic storms (another important cause of coral loss).

Although bleaching was first observed in 1983 (Glynn, P. W. 1984. Widespread coral mortality and the 1982-83 El Nino warming event. Environmental Conservation 11:133–146), the first truly global bleaching event occurred in 1998 when the atmospheric $CO_2$ concentration(ppm) was 366 and global mean surface warming was ~0.7C. This mass bleaching resulted in significant coral mortality globally at a Degree Heating Week threshold (DHW, a measure of the duration of a MHW) of 8-12. Since then, up to 71% of the world's reefs have experienced recent bleaching (Virgen-Urcelay & Donner 2023). But with repeated events, loss of sensitive corals and acclimation and adaptation, the DHW threshold has shifted but uncertainty remains with various authors arguing between 8-12 DHW as a critical threshold.

Thermal stress driven by increasingly warmer ocean temperatures, compounded by El Niño heating events, is the primary stressor of regional scale mortality of hard corals, Hughes et al. 2017; Houk et al. 2020, UNEP 2020; IPCC 2022). Heat stress results from small increases (1–2 °C) in seawater temperature above the summer maxima to which corals are acclimatised, destabilising the symbiosis between host corals and their symbiotic algae.

The first truly global bleaching event occurred in 1998, at ~0.7C global mean surface temperature and 366 ppm $CO_2$. This mass bleaching produced significant coral mortality globally at a threshold of 8-12 Degree Heating Weeks (DHW, calculated by the increase and its duration in weeks within a 12-week window). Observations indicated that up to 71% of the world's reefs have experienced recent bleaching (Virgen-Urcelay & Donner 2023). But with repeated events, loss of sensitive corals and acclimation and adaptation, the DHW threshold has shifted but uncertainty remains with various authors arguing between 8-12 DHW as a critical threshold.

Tipping points that have already been reached in seven ocean systems include bleaching of tropical coral reefs (IPCC 2022 Figure FAQ3.31). More than 80% of coral reefs are expected to experience annual severe bleaching by the middle of the century, even assuming 2°C of adaptation (UNEP 2020). Investigations have highlighted consequences of different levels of



warming (mostly not considering co-occurring/interacting stressors or the additional warming resulting from ocean warming
response to atmospheric $CO_2$ concentrations):

0.7°C - "In the late 1990s when global warming was around 0.7°C large-scale coral reef bleaching also became apparent …
supporting the lower boundary for this transition in respect of coral reefs" (Veron et al 2019; IPCC, 2022)
1.0°C - "temperatures of just 1°C above the long-term summer maximum … over 4–6 weeks are enough to cause mass coral
bleaching … and mortality (very high confidence)" (Hoegh-Guldberg et al 2018; Skirving et al 2019).
1.2°C - "Warm water (tropical) coral reefs are projected to reach a very high risk of impact at 1.2°C …, with most available
evidence suggesting that coral-dominated ecosystems will be non-existent at this temperature or higher (high confidence). At
this point, coral abundance will be near zero at many locations and storms will contribute to 'flattening' the three-dimensional
structure of reefs without recovery, as already observed for some coral reefs (Alvarez-Filip et al., 2009)." (Hoegh-Guldberg et
al 2018). Coral reef bleaching tipping points have already been reached in seven ocean systems (IPCC 2022).
1.5°C - "...coral reefs… will undergo irreversible phase shifts due to marine heatwaves with global warming levels >1.5°C
and are at high risk this century even in <1.5°C scenarios that include periods of temperature overshoot beyond 1.5°C (high
confidence)." (IPCC 2022). Projections predict 70-90% coral loss at 1.5°C (Hoegh-Guldberg et al 2018; IPBS 2019; Souter et
al 2021; Armstrong McKay et al 2022), whereas finer scale modelling projects a 95-98% loss (Kalmus et al (2022) and suggest
99% loss Dixon et al 2022).
2.0°C -: "literature since AR5 has provided a closer focus on the comparative levels of risk to coral reefs at 1.5°C versus 2°C
of global warming … reaching 2°C will increase the frequency of mass coral bleaching and mortality to a point at which it
will result in the total loss of coral reefs from the world's tropical and subtropical regions." (IPCC 2018). Predictions show
99% coral loss at 2.0C (Hoegh-Guldberg et al 2018; IPBS 2019; Souter et al 2021; Armstrong McKay et al 2022). Finer scale
modelling projects 100% loss at 2.0°C. (Dixon et al 2022; Kalmus et al 2022).

Ocean warming response times mask the impact severity of stated $CO_2$ and temperature levels. When overshoot is considered,
lower temperatures can have similar impacts to higher, with little difference in coral survival between an overshoot scenario
that peaks at 2°C and subsequently reduces temperatures to 1.5°C versus a 2°C scenario without a subsequent reduction in
temperatures (Tachiiri et al., 2019).

A centennial-scale index of extreme marine heat for the global ocean confirms the normalisation of historical heat extremes
with 2014 being the first year to exceed the 50% threshold extreme heat thereby becoming normal (Tanaka and Van Houtan
2022). The compounding heat stress of El Niño events on corals (Claar et al 2018; Hughes et al 2018; Lough et al 2018) may
increase with more frequent El Niño events linked with projected Arctic sea ice loss (Liu et al 2022; Kennel et al 2020; Kim
et al 2020) and Antarctic sea ice loss (England et al 2020).  Regardless of the projected heating impacts, real world observations



from the NOAA coral reef watch program demonstrates that coral reef damage is accelerating and underscores the threat
anthropogenic climate change poses for the irreversible transformation of these essential ecosystems (Eakin et al 2022).

## 4. Ocean acidification

Ocean acidification (OA) is the process of the increasing absorption of atmospheric $CO_2$ by the surface seawaters of the oceans,
which in turn reduces the calcification rates of most scleractinian tropical and subtropical corals (Comeau et al. 2014, Kornder
et al. 2018), and can alter the photo-physiology and calcification physiology of some corals (Comeau et al. 2018).
OA causes declines in coral calcification rates in laboratory simulations of future seawater (Comeau et al. 2018). Early work
predicted large-scale loss of coral calcification at catastrophic levels, whereby OA was projected to result in coral bleaching
and in some cases net dissolution of corals (see data within Leung et al. 2022). Contemporary research demonstrates that some
corals are resistant to OA (Comeau et al. 2018, Kornder et al. 2018). The most comprehensive modelling estimates are that by
year 2100 coral calcification would decline by 1% under RCP2.6, 4% under RCP4.5 and 15% at RCP8.5 (Cornwall et al.
2021). When combined solely with the metabolic effects of temperature increases, this decline would be 1% (RCP2.6), 8%
(RCP4.5), and 33% (RCP8.5). However, the calcification rates of susceptible coral taxa (e.g., *Acropora* spp.) would decline
by much more, and resistant species (e.g., *Pocillopora* spp. or *Porites* spp. generally) could be unaffected.
The direct metabolic impacts of OA do not manifest a tipping point, but tipping points at ecological levels are likely. The
negative impacts on coral and coralline algal calcification are direct negative effects, when combined with the direct positive
effects on other taxa (such as opportunistic turfing algae). Susceptible species would start to give way to tolerant species over
time (as generally occurs at natural analogues in the field Fabricius et al. 2011, Comeau et al. 2022), and other non-coral taxa
would start to dominate space on what once were traditional coral reefs. OA acts to alter the internal chemistry of corals and
coralline algae, slowing calcification rates. Species that are capable of maintaining stable internal carbonate chemistry or
compensate for these changes tend to be more tolerant to OA. However, of greater immediate importance to the majority of
corals will be successive marine heatwaves that will reduce the coral cover of less heat tolerant species, populations and
genotypes over the majority of the oceans in the near future (van Hooidonk et al. 2014, Cornwall et al. 2021, Logan et al. 2021,
Cornwall et al. 2023). Survivors of this evolutionary force will not necessarily be those that are tolerant to OA also, and thus
numerous tipping points in time could occur. Extensive meta-analysis of the impacts of ocean warming (Cornwall et al. 2019)
and ocean acidification (Cornwall et al. 2022) on coralline algae reveal that ocean acidification is likely a major threat to these
taxa which help bind reefs together. However, more work is required to understand whether there is a tipping point in the
important role they play on coral reefs.



## 5. Deoxygenation

Deoxygenation on coral reefs is the least studied of the climate change 'triple threat' that also includes warming and acidification (Hughes et al. 2020). However, there is sufficient evidence to say that dissolved oxygen is a critical resource on coral reefs, and that oxygen limitation (i.e. hypoxia) results in non-linearities and feedbacks that contribute to ecological tipping points (TPs) (Nelson and Altieri 2019). The consequences of crossing these TPs are perhaps most dramatically evident in sudden mass mortality events, which has led to calls to accelerate the research agenda on deoxygenation on coral reefs (Altieri et al. 2017).

The oxygen concentration threshold at which corals lose their ability to maintain homeostasis is 2 mg/L with lethal doses between 0.5-2 mg/L (Johnson et al. 2021a, Hughes et al. 2022). (See table). Coral reefs are vulnerable to a number of feedbacks that exacerbate deoxygenate events when TPs are exceeded. These include bleaching (Altieri et al. 2017, Johnson et al. 2021a,b, Alderdice 2021), excessive dead material from mass mortality events (Simpson et al. 1993), coral disease and algal growth (Dinsdale and Rohwer 2011), and shifts in the coral microbiome (Howard et al. in press).

The problem of deoxygenation on coral reefs is becoming more prevalent and severe in the Anthropocene from a combination of global climate change (Altieri and Gedan 2015, Pezner et al. 2023), as well as local pollution in the form of excess nutrient and organic matter (Diaz and Rosenberg 2008), that are magnified by local oceanographic patterns (Adelson et al. 2022). Two different methods independently estimated that 13% of coral reefs globally are at risk of deoxygenation, and the percentage of reefs that cross the threshold into this risk category is likely to increase with continued climate change (Altieri et al. 2017, Pezner et al. 2023).

Climate-related variables of temperature and acidification are also likely to exacerbate deoxygenation by affecting the physiological responses of corals and other reef organisms. It is widely recognized that increased temperatures lead to increased metabolic demand and decreased tolerance thresholds in marine organisms including corals (Vaquer-Sunyer and Duarte 2011, Alderdice et al. 2020, Weber et al. 2012). Given the prevalence, co-occurrence, and synergistic effects of these co-stressors with deoxygenation, a multi-stressor perspective is essential, and many of the assumed thresholds for TPs on coral reefs based on single or even double stressor treatments under laboratory experiments are likely overly conservative estimates.

We suggest that evidence to date for feedbacks and non-linear thresholds indicates that a TP framework should be used to guide future research on deoxygenation in coral reefs, and that hypoxia should be considered in studies of thermal stress and acidification.



## 6. Storm intensity

Tropical storms can temporarily reduce thermal stress (IUCN 2016; Bowden-Kerby 2023) but can also physically damage reefs. Ocean warming may increase the severity of cyclones (IPCC 2021; Setter et al 2022) and coral bleaching has likely reduced the ability of reefs to recover from cyclone damage (IUCN 2016). The likelihood of more intense cyclones within time frames of coral recovery by mid-century poses a global threat to coral reefs and dependent societies (Cheal et al (2017). The direct force of wind and waves, along with changes in storm direction, increase risks of physical damage and exposure to reduced water quality and sediment runoff (IPCC 2018). Storms contribute to unstable rubble substrate, compromising coral settlement (Sheppard et al 2020). Furthermore, frequent intense storms can hinder reef recovery (Puotinen et al., 2020). Setter et al (2022) ascribe a co-occurring stressor variable suitability threshold value of strength category <4 with a return time of >5 years (see table).

## 7. Sea level rise

Moderate rates of sea level rise (SLR) may potentially aid some reefs contend with thermal stress and thus have an antagonistic effect (Brown et al 2019; Cinner et al 2015; Baldock et al 2014). However, SLR rate and magnitude predictions (eg. Ciraci et al 2023, Vernimmen and Hooijer 2023) imply increasingly synergistic impacts, especially in the tropics (Hooiler and Vernimmen 2021; Cazenave et al 2022; Spada et al 2013). In addition to reefs drowning from exceeding *Darwin Point* thresholds (Grigg 2008) sea level rise can result in greater sedimentation and erosion stress (Laffoley et al 2016; Parry et al 2018; Williams NOAA 2019; Knowlton 2001). Saunders et al (2016) make the important point that while individual corals may keep pace with SLR, likely maximum reef framework accretion rate on reef flats is only 3mm yr$^{-1}$. Saintilan et al (2023) estimate likely vulnerability to RSLR at 7mm yr$^{-1}$ for coral reef islands. GMSL between 2006 and 2018 increased to 3.7 (3.2 to 4.2) mm yr-1 (IPCC 2021). Under SSP1-2.6, due to the risk of loss of reef structural integrity and transitioning to net erosion by mid-century the rate of sea level rise is very likely to exceed that of reef growth by 2050, absent adaptation (IPCC 2022). Depending on reef type and location SLR threshold rates range from 4-9mm yr-$^{1}$.

Closely connected seagrass and mangrove ecosystems (Guannel et al 2016; Earp et al 2018) are very vulnerable to projected SLR rate and magnitude (Saintilan et al 2023; Törnqvist et al 2021; Breda et al 2020; Sweet and Park 2020; Saunders et al 2014) which will further compromise coral reef resilience and functionality. In summary, SLR rate and magnitude looks increasingly likely to overwhelm the accretion ability of coral reefs which will be further challenged by increased wave energy, sedimentation, turbidity and resultant compromised light conditions for symbiont photosynthesis.



## 8. Pollution

Here we use pollution as an all-encompassing term covering sediment, eutrophication, turbidity and chemicals. Sedimentation reduces water clarity and hence energy supply, at the same time sediments settling on corals require greater energy to remove. It is caused mainly by land-based activities such as coastal urbanisation, with plumes travelling many km from disturbance sites (Brodie et al 2012). Organic pollution from sewage and agricultural run-off (e.g. fertiliser) are the main causes of eutrophication, which reduce light, actively poison invertebrates, introduce pathogens and reduce resistance to disease with direct impact on corals being decreased colony sizes, growth anomalies, and reduced growth and survival (Setter at al 2022). Metals and organic chemicals can rupture cell membranes, disrupt enzyme pathways reducing corals' ability to resist other stressors. Plastics have also been identified as another major cause of coral reef stress due to light interference, toxin release, physical damage, anoxia and increasing the likelihood of pathogen disease 20-fold (Lamb et al.2018).

## 9. Disruption

Here we are using disruption as a term covering land use change, human population density and overfishing. Land use can be used as a proxy for quantifying land-based pollution and other human stressors on coral reefs (Packet et al 2008, Cinner et al 2012, Setter et al 2022). To calculate reef change threshold exceedance, Setter et al (2022) use an ideal value of summed proportion agricultural/urban land use <0.5 in a 50km radius around a reef. Setter et al (2022) use human population density as the closest indicator available to quantify local human stressors, involving coral growth anomalies and disease, low biodiversity and fish biomass and reduced growth and survival. Perhaps the most direct physical human impact is overfishing with IPBS stating that more than 80% of the world's coral reefs are severely over-fished or have degraded habitats (McClanahan et al., 2015), which disrupts ecosystem balance leading to uncontrolled algal growth and dominance.

## 10. Disease

Diseases are major drivers of the deterioration of coral reefs and are linked to major declines in coral abundance, reef functionality, and reef-related ecosystems services (Alvarez-Filip et al 2022). Disease outbreaks are posing severe consequences for coral reef ecosystems, resulting in extensive coral mortality and endangering their long-term survival. Noteworthy events include the rapid proliferation of diseases like Stony Coral Tissue Loss Disease (SCTLD) (Alvarez-Filip et al 2022), black band disease (BBD), and various forms of white syndrome. Regions such as the Great Barrier Reef, the Caribbean, the Pacific Islands, and the Indian Ocean have been particularly impacted by these outbreaks, in some places surpassing the devastating impact of bleaching events by causing even greater coral mortality. Coral diseases stand out as being driven largely by a changing environment and are contributing to whole ecosystem regime shifts (Thurber et al (2020). Viral infections of coral symbiotic dinoflagellate partners (Symbiodiniaceae) will likely increase as ocean temperatures





continue to rise, potentially impacting the foundational symbiosis underpinning coral reef ecosystems (Howe-Kerr et al.
284  (2023).

## 11. Invasive species

Increased native and invasive coral predator and competitor populations can have severe impacts on reef ecosystems. A prime
example is the impact on the Great Barrier Reef by the crown-of-thorns-seastar (COTS) the outbreaks of which are attributed
to a combination of increased larval survivorship due to higher food availability, linked with anthropogenic runoff and warmer
sea temperature facilitating faster settlement of larvae (Uthicke et al 2017). The coral-killing sponge, *Terpios hoshinota* is a
global invasive species which has led to a significant decline in living coral cover at various geographical locations (Thinesh
et al 2017).

## 12. Stressor interactions

Some studies find an antagonistic interaction between multiple stressors (Darling et al., 2010; Ellis et al., 2019; Johnson et al.,
2022). However, a wide variety of interacting and predominantly synergistic stressors have been found to co-occur
(Ateweberhan et al 2013; Boyd et al 2018; Bijma et al 2013; Ellis et al 2019; IPBS 2019; Zscheischler et al 2018; (IPBS 2019);
ICRS 2021; IPCC 2022; Setter et al 2022), generally lowering the thermal threshold for bleaching and/or mortality, bringing
forward timing of collapse, or even surpassing thermal stress in local importance (e.g. overfishing, disease, pollution,
invertebrate predators; ocean acidification) (Anthony 2016, Ban et al. 2013; Cramer et al. 2020; Darling et al. 2019; Edmunds
et al. 2014; IPBS 2019; Rocha et al. 2013; Setter et al. 2022; Veron et al 2009). An increase in reefs facing 'unsuitable
conditions' from 44% in 2005 to, under worst case scenarios, 100% by 2055 under any one of several stressors, by 2035 for
cumulative stressors under RCP8.5 (Setter et al. 2022).

## 13. Reef impact example

### 13.1 Chagos Archipelago demonstrates positive feedback (tipping points).

Observations from the Chagos Archipelago, central Indian Ocean, reveal several related lessons. Coral cover collapsed after
the heatwaves of 2015-2016 by 90%. There were very few surviving adults capable of spawning, with survivors likely
weakened and observations showed about three years was needed before they recovered sufficiently to recommence growth
(Sheppard and Sheppard 2019).
Settlement of larvae, when it occurred, was compromised due to disintegrating substrates. In many shallow areas, where wave
energy had already swept the substrate clear of rubble, large areas are becoming covered by the encrusting and bioeroding
sponge *Cliona* spp (Sheppard et al 2020 skeletons formed a very abrasive layer on the substrate and, like liquid sandpaper,




almost no larvae were seen in these areas. These sponges are clearly increasing; with one reef showing over 80% Cliona cover
preventing coral larvae settlement.
On at least one lagoon floor, the former foliaceous coral dominance was also killed with skeletons disintegrating resulting in
fine sediment covering all surfaces. Both sedimented surfaces and turbid water are hostile to larval settlement, and none were
seen in such areas over many hectares.
The scenario of fewer corals producing fewer larvae, more turbid water in some areas and less substrate available for settlement
is a classical positive feedback or tipping point situation. These factors all act synergistically in a direction that inevitably leads
to an ever more impoverished reef system. Recovery from this will require a prolonged period without heat stress and a gradual
removal of the vast volumes of sediment and rubble left from previous bleaching events.

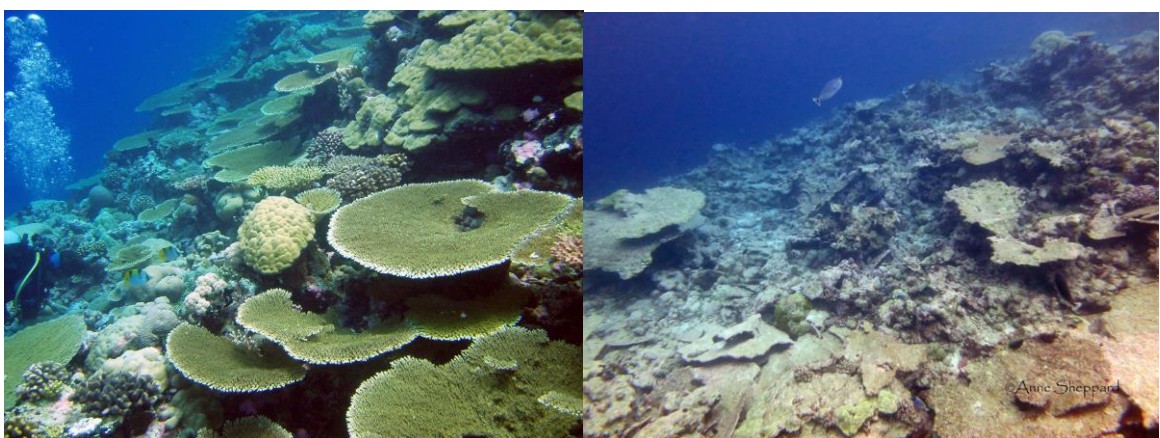


**Figure 1: Reef slope on Salomon atoll, Chagos Archipelago, before and after the mass mortality caused by warming in**
**2015**

## 14. Cascade effects contributing to coral reef tipping point threshold sensitivity

Accelerating West Antarctic Ice Sheet melt (Naughten et al 2023), increasing methane emissions (Zhang et al 2023) and Arctic
sea ice decline have the potential to increase rate and magnitude of coral reef stressor impacts. For example, Liu et al (2022)
predict that 37–48% of the increase of strong El Niño near the end of the 21st century is associated specifically with Arctic
sea-ice loss.

## 15 Conclusion

Mass coral mortality repeated more than twice per decade and over local, regional and ocean scale, and by aggregation to
global scales, is increasingly recognized as giving insufficient time for recovery of impacted populations and ecological



function (Hughes et al. 2018a, 2018b, Obura et al. 2022). Ecological and biogeographical (spatial) feedback loops prevent
recovery through failure of reproduction, dispersal, recruitment and growth of corals (Sheppard et al 2020) (see box x).

Other stressors reduce the ability of corals to resist thermal stress thus lowering tipping thresholds. Increasing frequency and
intensity of regional scale coral mortality events (1+ °C warming) are suggestive of the majority of coral reefs already having
reached a bleaching tipping point (IPCC 2022). The potential for thermal refuges for corals under likely future scenarios is
doubtful (Beyer et al. 2018; Dixon et al. 2022; Setter et al. 2022) as very few or no reef areas are predicted to remain below
tipping thresholds of all key stressors. The existence of putative refuges at greater depths (Bongaerts and Smith 2019) or higher
latitudes (Setter et al. 2022) are not strongly supported by recent work (Hoegh-Guldberg et al 2017; IPCC 2018; IPCC 2022).
**15.1 Tipping thresholds**
Veron et al (2009) states 'when $CO_2$ levels reached ∼340 ppm (with water temperatures reflecting a 10-year time-lagged
response to <∼326 ppm) sporadic but highly destructive mass bleaching occurred in most reefs world-wide, often associated
with El Niño events. At the 2009 $CO_2$ level of 387 ppm, allowing a lag time of 10 years for sea temperatures to respond, most
reefs world-wide are committed to an irreversible decline with eventual annual bleaching. If $CO_2$ levels reach 450 ppm
(expected to occur by 2030-240), allowing a lag time of 10 years, reefs will be in rapid and terminal decline world-wide from
multiple synergies arising from mass bleaching, ocean acidification, and other environmental impacts and will cease to have
most of their current value to humanity. Veron et al concluded that to ensure the long-term viability of coral reefs, atmospheric
$CO_2$ levels must be reduced significantly below 350ppm. Considering subsequent GHG emission trajectories, overshoot
magnitude and ocean warming response times, the 350ppm $CO_2$ threshold could now be considered optimistic but, pending
further analysis, this remains the best available $CO_2$ threshold value, with a suggested range of 326-400 ppm.

The recent Global Tipping Points Revision initiative focused on temperature tipping point thresholds and suggested that 'the
critical threshold of 1.5°C (range 1-2°C) (Armstrong McKay et al. 2022) should be adjusted, narrowing and lowering the range
to 1-1.5°C, with a middle estimate of 1.2°C, marked by the multi-year global coral reef bleaching events of 2015-2017 (IPCC
AR6 WG2 Ch3 2022; IPCC SR1.5 Ch3, 2018; Dixon et al. 2022; Setter et al. 2022). The co-occurrence of additional synergistic
drivers also support lowering the critical threshold (Willcock et al. 2023) and there is evidence of accelerating collapses at
increasing spatial scales (Cooper et al. 2020). The combined effects of long-term warming, sea level rise, ocean acidification,
deoxygenation, and other stressors, bears more investigation to identify the lower critical threshold for the coral reef tipping
point.'

We recognise the warming thresholds of 350ppm $CO_2$ (with a suggested range of 326-400 ppm) and 1.2°C (with a suggested
range of 0.7-1.5°C) whilst acknowledging that interacting stressor impacts, ocean warming response time, GHG emissions
overshoot and cascade effects have yet to be robustly assessed. These and other uncertainties around tipping point sensitivities





for such a crucially important ecosystem underlines the imperative of robust assessment (Heinze et al 2017; Aronson and
Precht 2016) and, in the case of knowledge gaps, employing a precautionary principle (OECD 2022; Rockström et al 2023) to
tipping points and favour lower range threshold value. The key take home message is that due to lags in the climate system
and interactions with other stressors, we've very likely already crossed the tipping point for coral reefs. Without climate
mitigation action to realise the necessary temperature and GHG concentration levels our remaining chance to reverse this
situation will be lost. Recognising threat severity is essential if the necessary response action is to be realised.
**Table 1: Stressors on coral reefs, their interactions and tipping points.** Stressor categories are defined in the text above.
Arrows indicate synergistic interactions, indicating greater impacts due to interactions. Data from this table is based on
Ateweberhan et al (2013). Tipping points are summarised from the text above. Rate column indicates the pace of impact of
stressors on reef habitats.

| Stressor | Impact Rate | Impact Scale | Tipping Point Threshold | Notes |
|---|---|---|---|---|
| ocean warming | Progressive | Global | 1.2°C (0.7-1.5°C) 350ppm $CO_2$ (326-400ppm) | Warming oceans make heatwaves more likely, increase storm intensity, create sea-level rise through thermal expansion. Warming induces bleaching increasing disease risk, lowering calcification. (Bak et al Burke et al 2023, Eakin et al 2008, Marshall & Clode 2004, Rosenberg & Ben-Haim 2002, Ward et al 2007; Veron et al 2029). |
| marine heatwaves | Abrupt | Regional | 8-12 DHW | Heatwaves induce bleaching increasing disease risk, lowering calcification (Miller et al 2009). |
| storm intensity | Abrupt | Regional | category 4 with a return time of 5 years | More storms lead to more sediment resuspension. Strong storms are linked to fragmentation and considerable damage to reefs. High frequency of such storms prevents recovery. Storms can reduce sea temperatures and save bleaching coral from mortality (Gardner et al 2005, Manzello et al 2013, Carrigan & Puotinen, 2014, Puotinen et al 2020, Setter et al 2022). |
| ocean acidification | Progressive | Global | 538-572 $CO_2$ | Reduced calcification increases disease risk, weakened skeletons are vulnerable to storms (Setter et al 2022, Anthony et al 2011, Portner 2008, Suwa et al 2010, Steffen et al 2015). |
| sea level rise | Progressive | Global | 8mm $yr^{-1}$ SLR (4-9 mm $yr^{-1}$) | Deeper, cooler water potentially reduces thermal stress (Brown et al 2019; Baldock et al 2014). High SLR rate and magnitude transitions from antagonistic to synergistic effects, reducing light availability, increasing sedimentation and turbidity (Laffoley et al 2016; Perry et al 2018; IPCC 2022). Synergistic mangrove and seagrass impacts compound reef stress (Saintilan et al (2023). |
| deoxygenation | Progressive -> Abrupt | Local | 0.5-2 mg/L (mortality threshold) | Deoxygenation lowers the thermal threshold of coral bleaching (Alderdice et al 2022) and increases disease (Dinsdale and Rohwer 2011). |



| pollution | Progressive -> Abrupt | Local | 0.45 µg/L chlorophyll? | Pollution causes stress and increases disease risk. Eutrophication increases deoxygenation (Laffoley & Baxter 2019, Redding et al 2013, De'ath and Fabricius 2010). |
|---|---|---|---|---|
| disruption | Abrupt | Local | agricultural/urban land use <50% in a 50km radius around reefs | Overfishing can lead to algae overgrowth inducing disease & lowering calcification (Packett et al 2009, Maina et al 2013, Prouty et al 2017, Kroon et al 2014, Fabricius 2005) |
| disease | Abrupt | Local to regional | Roughly 1-2°C above ambient seasonal highs. | Some coral diseases (but not all) have been linked to both marine heat waves and the longer-term warming trend (Bruno et al. 2007, Randall and van Woesik 2015). In the Caribbean, coral disease has been the primary cause of coral loss. |
| invasive species | Progressive -> Abrupt | Local | species dependant | Invasive species predation scars leave corals susceptible to disease (Nicolet et al 2018). |

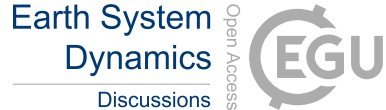

**Figure 2: Visualisation of stressor interactions. Red links denote synergistic associations (expanding negative impacts) and blue links denote antagonistic associations (one factor ameliorating the impact of another).**

**Acknowledgements**

We are grateful to Richard Betts for advice on atmospheric $CO_2$ and temperature associations and ocean warming response times.



**Conflict of interest**

The contact author declares that none of the authors have any competing interests.

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
