# Peer review of "Assessment of warm-water coral reef tipping point thresholds"

_Earth System Dynamics, 2023_

## Author Comment (AC2)

Author response to reviewer comments for https://doi.org/10.5194/esd-2023-35 "Assessment of warm-water coral reef tipping point thresholds"

We welcome the attention already received for this manuscript in terms of website views and social media posts. We hope that this positive attention demonstrates the value of this review.

Reviewer 1 comments:

COMMENT 1: The paper is a comprehensive and reasonable review article focused on coral reefs under local as well as global environmental changes, which would be valuable for researchers and managers in this field.

*Author response: We thank the reviewer for this positive assessment.*

COMMENT 2: However, while the paper describes the concept as a threshold characterized by non-linear hysteresis or cascading effects resulting from positive and negative feedback loops, it falls short in offering a systemic elucidation of why the outcome becomes synergistic rather than merely additional from the point of system science. To rectify this critical shortcoming, urgent measures should be taken to enhance the originality and academic contribution of this paper. Strongly recommended actions include incorporating more quantitative analyses or empirical evidence supporting the discussed concepts and delving deeper into the systemic dynamics driving the observed phenomena.

*It was our intention to create a review article that summarises our current knowledge of tipping points thresholds and dynamics for warm-water coral reefs. We feel this is beyond the scope of this paper, would change the nature of the article and move it away from a focussed review. We have changed the title to emphasise that this is a review article.*

*Our primary intention is to convey the diversity of contributing stressors and the need to consider their impact significance in threat assessments, rather than the interaction directions and strengths. We have included a large amount of high level references that provide insight into the kind of interactive dynamics and outcome information the reviewer's referring to.*

*Our intention is not to labour the synergistic elements of interactions, rather to demonstrate there are many stressors that require investigation/consideration to provide any comprehensive assessment of coral reef futures. We do clearly state in the manuscript that robust interactive stressor considerations have yet to be sufficiently included in tipping point assessments. A key aim of our manuscript is to highlight important stressor considerations and to encourage their robust inclusion in assessment initiatives.*

COMMENT 3: In conclusion, the lack of an effective explanation due to the absence of original data significantly undermines the academic value of this paper. Consequently, the paper, in its current state, is more suitable for publication in a general introduction book rather than an academic journal.

*The catalyst and main reference base for this manuscript is the global tipping points report (Lenton 2003) and seeks to be a review informed by expert opinion. We leave it in the hands of the editors to decide on the merit of this work as a review article and its value in helping to advance this vitally important assessment area. We hope that the extensive revision of this manuscript that we are working on will largely address these comments.*

**COMMENT 4: Other specific comments:**

- **Lines 111-116 are mostly repeated in lines 124-129.**

*We have revised the manuscript to remove repetition and improve clarity.*

- **Chapters 8. Pollution, 9. Disruption, and 10. Disease should have their threshold values shown.**

*We have added threshold values for these and other stressors in the revised table, where there is information available. We have now included threshold values for all stressors except invasive / problem species.*

- **Chapter 11. Crown-of-thorns seastar is not an invasive species; it is common in coral reefs, but its problem lies in its outbreak.**

*We did not intend to suggest it was an invasive and the native predator context is made in the text but we will amend the section title (and Table 1 text) to better convey that both invasive and native 'problem' species are included in this section and ensure named species are correctly ascribed.*

**Reviewer 2 comments:**

**COMMENT 1**

The paper reviews climate impacts on coral reefs to consider adjusting the tipping points for coral reefs. It is not clear until the conclusions that they are suggesting a further reduction in temperature thresholds from the original 1.5 estimates. There are many problems with this assessment that are briefly outlined below. Perhaps, the greatest is that it is a poor review of the literature in that it is out of date and selective citation of the literature. I only give a small number of papers that should be considered in a thorough and up-to-date review. The authors needs to do a systematic review of the recent literature and write a current status review that is more balanced, quantitative, and explains the decisions better.

*We thank the reviewer for the suggestion of new articles. This is such an active area of research with new information coming out all the time, certainly since we submitted this manuscript. We are in the process of a substantial revision, including many new supporting references.*

There is also the problem of the tipping point definition and how it is measured and if they are rates or states, what is the time scale of the change, what are the specific irreversible changes, and what are the confidence intervals around estimates. There are two recent reviews that summarises these problems well and would cause a reconsideration of the confidence in this new lower threshold.

*We clearly state the IPCC tipping point definition and are using published values described by other authors as tipping points rather than creating new ones. In terms of timings, these values are in the context of ongoing climate change and the timescales of observed and predicted changes. We have chosen to include both rate and state related issues and have tried to clarify the text accordingly. All supportive references are provided to point the reader to their sources if further detail is required.*

Klein, S. G., C. Roch, and C. M. Duarte. 2024. Systematic review of the uncertainty of coral reef futures under climate change. Nature Communications 15:2224.

McClanahan , T. R. 2022. Coral responses to climate change exposure. Environmental Research Letters 17:073001.

*These references have been added to the revised manuscript.*

Table 1 is the core finding of the paper but it still reads as poorly articulated draft. I would suggest the authors review the current literature, summarize it and present table 1 as a key element. Then, most of the text can focus on the evidence or lack of evidence for the conclusions in this table. The authors could conclude with what are the key unknowns required to better estimate the tipping points. Otherwise, the current draft is an uncritical, out of date, and selective citation review of a much richer, data intensive, and nuanced coral reef literature that should be used to make these important conclusions.

*We have substantially revised Table 1 removing the interactions to another section of the manuscript to make this a clearer summary of published tipping points. We have updated references to try to bring this manuscript as up to date as is practically possible in line with the scope and intent of the manuscript.*

Abstract

The abstract is diffciult to distinguish from similar conclusions of past work. It is not that clear or adding much to past work from my reading as the context of the works is missing.

*The abstract has been revised (along with all the manuscript).*

What a tipping point threshold is should be defined early, otherwise the text is mysterious. I assume this is an irreversible change but that often depends on the time scale of observation and what is being measured. Readers will want to know.

*In addition to the tipping point threshold definition in the introduction section, we have now included a brief description in the abstract. These definitions are drawn from IPCC and the recent tipping points review.*

Usually, a sentences on methods for assessing tipping points is required to understand the results. It should be stated that this is a review.

*As stated above this is not presenting new analysis, we have amended the title, abstract and main text to make it clear that this is a review. Including methodologies for every stressor is beyond the scope of this work, but supporting references are provided in the table if readers wish to delve deeper into methods. We also highlight the importance of robust inclusion of stressor interactions and their combined significance in future assessments. Doing so will better enable individual and interacting stressor influences to be determined/constrained. We consider this to be an urgent message/communication remit of the manuscript.*

The suggested tipping points seems similar to previous work, is it different and by how much? What would cause it to change.

*This review will necessarily tread ground that others have already covered, our objective here is to bring a wide range of information into a single document. We believe this is a useful exercise that may be of value for future assessment efforts. The outcome of the global tipping points review*

*indicated that assessments of tipping points need further consideration so they could change in the light of ongoing & further robust analysis. In particular we highlight the importance of including the wider range of stressor considerations detailed in the manuscript.*

 What are the uncertainties.  How were they addressed?

*The revised text delves further into uncertainties around the published thresholds and stressor interactions and it is part of the purpose of this work to encourage others to explore these uncertainties further and in the meantime apply precautionary principles to their approach.*

**Introduction**

**It is important to distinguish states from rates.  Rates determine states but this is not clear and confused as written. The states are also vague, is is coral cover, calcification, ecological state and integrity, biodiversity, etc?**

*The text has been significantly re-written to convey the significance of both rates (of change) and states (i.e. currently observed values) and provide more information on each to help  draw out the relevancy of both.*

**There is a much larger literature than is being cited here and the introduction seems to out of date given the many recent meta-analyses and reviews. I would suggest the authors look closer at the literature in the past 5 years and mostly cite these large data compilation studies. Much is being learned and it is not always in line with the general consensus that is being expressed in this paper. These are just a few examples that come to mind.**

 *We have substantially revised the text to incorporate more references (see comments above), including those kindly provided by the reviewer.*

 **It is worth remembering that permanent reversal and large scale changes is not just due to climate but also overfishing on large scales. These have been shown to have tipping points and large scale changes. Here is a recent meta-analysis paper.**

**McClanahan, T. R., A. M. Friedlander, L. Wantiez, N. A. J. Graham, J. H. Bruggemann, P. Chabanet, and R. M. Oddenyo. 2022. Best-practice fisheries management associated with reduced stocks and changes in life histories. Fish and Fisheries 23:422 - 444.**

 *We highlight the importance of non-climate change stressor impacts and  thank the reviewer for the reference, we note that this paper does not include any explicit definition of tipping point or threshold values. We include  a section on "disruption" that includes fishing impacts and will include this reference therein.*

 **P103 – this threshold is not being well supported by recent studies. There are many large studies that do not find this to be very useful.  Here is just one.**

 **DeCarlo, T. M. 2020. Treating coral bleaching as weather: A framework to validate and optimize prediction skill. PeerJ 8:e9449.**

*While we recognise there is some conflicting opinion over the significance of ocean warming and heatwaves, we believe the statement that global heating is a main driver of coral declines is well supported by current research (independent of any particular threshold).*

If you are going to cite papers that use these threshold models, perhaps other approaches and findings should be cited as well. Low oxygen has been shown to be associated with higher coral cover, so the theory here is not supported empiricallly.

Vercammen, A., J. McGowan, A. T. Knight, S. Pardede, E. Muttaqin, J. Harris, G. Ahmadia, Estradivari., T. Dallison, E. Selig, and M. Beger. 2019. Evaluating the impact of accounting for coral cover in large-scale marine conservation prioritizations. Diversity and Distributions 25:1564-1574.

McClanahan, T. R., and M. K. Azali. 2021. Environmental variability and threshold model's predictions for coral reefs. Frontiers in Marine Science 8:1774.

*We agree that dissolved oxygen is a critical factor for coral reefs, and the Vercammen et al. 2019 and McClanahan & Azali 2021 papers confirmed that with their analysis of the Bio-ORACLE database. We maintain that (1) oxygen can become limiting for corals, (2) that oxygen limitation can affect diversity and cover, and (3) that oxygen limitation will be observed below a threshold or critical value where corals are no longer able to maintain metabolic homeostasis (commonly <1-2 ml/l) (e.g., Altieri et al. 2017, Johnson et al. 2021).*

*The range of dissolved oxygen values observed in McClanahan & Azali 2021 are ~4.35-4.85 ml/l (figure 2) which are concentrations well above levels found to be limiting in experimental trials and field observations with corals. The information available in the Vercammen et al. 2019 paper and online supporting information were not adequate to determine the values included in their analysis, but based on the WOD09 data availability and their interpolation across available data points, they are likely to have integrated spatially and temporally in such a way that would miss the extreme values (i.e., below critical thresholds) associated with deoxygenation events that can be detrimental to coral cover and diversity.*

*The relationship between dissolved oxygen and hard coral is both positive and negative in McClanahan & Azali 2021 depending on the interval of oxygen values within the relatively narrow range presented in fig 2. There is insufficient information presented in Vercammen et al. 2019 to determine whether the relationship is negative, positive, or both – only that there is a significant relationship.*

*So there does not appear to be sufficient evidence that low oxygen (<1-2 ml/l) is associated with high coral cover. Further, it should be noted that oxygen CAN be a limiting factor for corals below certain thresholds, but that at higher ranges it could likely be serving as a proxy for other factors related to ecosystem metabolism (e.g., carbon dioxide, pH, photosynthesis), or could be limiting for other organisms that interact with corals (e.g., corallivores) even though it is not limiting for the corals, or a product of the abundance of primary producers that influence coral abundance and health. And so the relationship between oxygen concentration and coral abundance and diversity likely requires additional ecological and scale-dependent information for proper interpretation.*

*Altieri, A.H., Harrison, S.B., Seemann, J., Collin, R., Diaz, R.J. and Knowlton, N., 2017. Tropical dead zones and mass mortalities on coral reefs. Proceedings of the National Academy of Sciences, 114(14), pp.3660-3665.*

*Johnson, M.D., Swaminathan, S.D., Nixon, E.N., Paul, V.J. and Altieri, A.H., 2021. Differential susceptibility of reef-building corals to deoxygenation reveals remarkable hypoxia tolerance. Scientific Reports, 11(1), p.23168.*

There is a view that sea level rise will increase calcification but this was largely developed before the views of coral mortality.

*We thank the reviewer for this comment. We believe this relates to the potential increase in habitat availability in certain circumstances, which we reference in the current manuscript where we consider the antagonistic and other interactions of sea level rise with other factors. We will keep these comments in mind as we revise the section on sea level rise.*

There are also the sediments and seabird studies that show increased nutrients improve reef condition and resilience.

Graham, N. A., S. K. Wilson, P. Carr, A. S. Hoey, S. Jennings, and M. A. MacNeil. 2018. Seabirds enhance coral reef productivity and functioning in the absence of invasive rats. Nature 559:250.

MacNeil, M. A., C. Mellin, S. Matthews, N. H. Wolff, T. R. McClanahan, M. Devlin, C. Drovandi, K. Mengersen, and N. A. J. Graham. 2019. Water quality mediates resilience on the Great Barrier Reef. Nature Ecology & Evolution 3:620.

*We thank the reviewer and will investigate this further and ensure it is captured in the appropriate section. We note here that the current biodiversity crisis extends beyond corals to include seabird declines that may negatively impact coral reefs.*

L353 – the authors should give the reasoning for lowering the thresholds not just cite papers. The recent review by Klein et al 2024 indicate that lack of confidence intervals around thresholds.

*We provide verbatim reasoning for the thermal and $CO_2$ thresholds derived from the conclusions of the global tipping point revision (Lenton et al 2023). We also provide ranges for thresholds where these are available. We also address confidence considerations, including the rationale for better inclusion of interacting stressors in the assessment process and the need to adopt a precautionary approach when considering confidence uncertainties. We consider these to be important messages of the manuscript.*

Much of table 1 is out of date. However, they do cite Veron et al. 2029 to compensate for this problem. I look forward to reading it in 5 years. Authors should know the literature and how to organize and write a review and subsequently submit a more polished paper.

*We have updated our references and have corrected typographic errors. Please also refer to our table related replies above.*

---

## Referee Report (RR1)

**Referee Report 17ᵗʰ August 2024**

Considerations for determining warm-water coral reef tipping points.
Paul Pearce-Kelly, Andrew H. Altier, John F. Bruno, Christopher E. Cornwall, Melanie McField, Aarón Israel Muñiz-Castillo, Juan Rocha, Renee O. Setter, Charles Sheppard, Rosa Maria Roman-Cuesta, and Chris Yesson

**General comments:**

The manuscript is a very useful contribution outlining the potential tipping points arising from the many faceted issues, and interactions therein, that impact and increasingly threaten coral reefs.

More information on the rationale for selection of 1.2 C and 350 ppt $CO_2$, as per the Veron et al. (2009) paper, would be useful. The paper could benefit from an expanded Section 13 on Chagos, to include more of the history, given it is one of the best studied, remote reef systems. The paper could also benefit from additional case-studies from Caribbean (nutrient enrichment, over-fishing, heat-waves, diseases, invasive species) and Great Barrier Reef (COTS, storms, run-off, heat waves, etc.), with summary noting differences in resilience among the three systems in respect to oceanographic connectivity and variability therein, coral population sizes, habitat heterogeneity, depth ranges etc.

The paper would benefit from a hard edit to shorten some sections that appear repetitive; and with shifting of some text that may fit better in other sections. The Interactions sections may be better combined into one section following all those dealing specifically with the various stressors. This would help to minimize repetition.  Eg. Ocean acidification section also presently includes comments on heat waves – those could be covered solely in the Interactions section. I have suggested shuffling and shifting some text accordingly (see below Lines 196-210). Similarly in Disruptions section, some text seems to fit better in Pollution section.

**Specific comments:**

References could be improved by including more of the primary sources. Some examples are provided. Citations in text need to be standardized as per journal requirements.

**Lines 39-42:**

"They are also among the most sensitive ecosystems to anthropogenic driven stressors with an estimated 50% of global live coral cover having been lost over the last 50 years (Souter et al., 2021, WWF 2022), primarily due to ocean warming (and related climate change threats of ocean acidification and deoxygenation), but in some locations also due to fishing, pollution, and disease (IPCC 2022)."

LD: Loss of cover – consider noting '... albeit with significant temporal fluctuations, as on the Great Barrier Reef (see AIMS LTMP reports).

https://www.aims.gov.au/monitoring-great-barrier-reef/gbr-condition-summary-2023-24

Also, Crown-of-thorns seastar (*Acanthaster* spp., COTS) outbreaks have caused sig. loss of coral cover across the Indo-Pacific since at least the 1960s. Heat-wave driven bleaching mortalities are increasing rapidly, since 1998. However, to date, loss of significant cover from ocean acidification and deoxygenation is not well established (at least as far as I am aware),

although these are serious looming threats.  Nutrient enrichment (as a specific form of pollution) has had sig. impacts in the Caribbean and in parts of the Coral Triangle (eg. Areas of Java Sea).

**Line 63:**

"Approximately half the live coral cover on coral reefs has been lost since the 1870s …"

LD: presumably 1970s? Also recently estimated as dropping from 36 to 19 percent from 1997 to 2018 (Tebbett et al. 2023), with decreases most severe in the Western Atlantic and Central Pacific.

Tebbett SB, Connolly SR, Bellwood DR (2023) Benthic composition changes on coral reefs at global scales. Nat Ecol Evol 7: 71–81. https://doi.org/10.1038/s41559-022-01937-2

**Lines 67-70:**

"… regional die-offs (e.g. Western and Central Indian Ocean, Great Barrier Reef, Mesoamerican Reefs) (Le Nohaïc et al., 2017; Amir 2022; Muñiz-Castillo et al., 2019; Obura et al., 2022; Sheppard et al., 2020), with most reef regions having experienced multiple die-off events (Darling et al., 2019; Cramer et al., 2020; IPCC 2022). Coral reef bleaching tipping points have already been reached in seven ocean systems (IPCC 2022)."

LD: On GBR, Earth's largest and arguably oceanographically best-connected reef system, and elsewhere, the 'die-offs' (coral cover losses) since the 1960s (COTS mainly) have been interspersed with recovery of cover, as has again recently happened, but prior to severe 2024 mass bleaching.

https://www.aims.gov.au/monitoring-great-barrier-reef/gbr-condition-summary-2023-24

This is not to downplay the seriousness of present impacts and future risk, nor the shifts in community structure (eg. See Richards ZT, Juszkiewicz DJ, Hoggett A (2021) Spatio-temporal persistence of scleractinian coral species at Lizard Island, Great Barrier Reef. Coral Reefs 40: 1369-1378. https://doi.org/10.1007/s00338-021-02144-4), but to add some nuance to the statements.

**Line 100:**

"…is over centennial time (IPCC 2021).

**Lines 104-105:**

"(IPCC 2022). Overshoot of multidecadal time spans imply severe risks and irreversible impacts in many ecosystems Meyer et al. (2022), including …

**Lines 110-111:**

"Increasingly warmer ocean temperatures, driven by Anthropogenic climate change, compounded by El Niño heating events, is the primary stressor of regional scale mortality of scleractinian corals,.

LD: … and ocean-basin …

**Line 113:**

Primary sources would be better: Eg. Cite Barbara Brown and John Ogden, Peter Glynn, Ove Hoegh-Guldberg, among others, here.

Glynn, P.W., D'Croz, L. (1990) Experimental evidence for high temperature stress as the cause of El Niño-coincident coral mortality. *Coral Reefs* 8: 181–191. https://doi.org/10.1007/BF00265009

Brown, B.E. and Ogden, J.C. (1993) Coral Bleaching. *Scientific American* 268 (1): 64-70. https://doi.org/10.1038/scientificamerican0193-64

Glynn, P.W. (1996) Coral reef bleaching: facts, hypotheses and implications. Global Change Biology 2 (6): 495-509. https://doi.org/10.1111/j.1365-2486.1996.tb00063.x

Hoegh-Guldberg, O. (1999) Climate Change, Coral Bleaching and the Future of the World's Coral Reefs. Marine and Freshwater Research 50: 839-866. http://dx.doi.org/10.1071/MF99078

**Line 143:**

"Since the first bleaching event of 1998,…

LD: Pedantic, but this should be '… first global bleaching event …' Bleaching had been documented since at least 1960s following flooding (Tom Goreau Snr).

**Line 145-146:**

"With repeated events, loss of sensitive corals and acclimation and adaptation, the DHW thresholds may change (Lenton et al., 2023)

LD: Also see and cite:

van Woesik R, Kratochwill C (2022) A global coral-bleaching database, 1980–2020. Sci Data 9: 20. https://www.nature.com/articles/s41597-022-01121-y

van Woesik R, Shlesinger T, Grottoli AG, et al. (2022) Coral-bleaching responses to climate change across biological scales. Glob Chang Biol 28: 4229-4250. https://doi.org/10.1111/gcb.16192

Donner SD, Rickbeil GJ, Heron SF (2017) A new, high-resolution global mass coral bleaching database. PLoS One 12:e0175490. https://doi.org/10.1371/journal.pone.0175490

LD: The time series suggested a possible increase in coral thermal tolerance – see Virgen-Urcelay and Donner 2023, Shlesinger and van Woesik 2023.

Shlesinger T, van Woesik R (2023) Oceanic differences in coral-bleaching responses to marine heatwaves. Sci Total Environ 871: 162113. https://doi.org/10.1016/j.scitotenv.2023.162113

Virgen-Urcelay A, Donner SD (2023) Increase in the extent of mass coral bleaching over the past half-century, based on an updated global database. PLoS ONE 18: e0281719. https://doi.org/10.1371/journal.pone.0281719

**Lines 182-184:**

The introduction to ocean acidification could benefit from a more detailed explanation of the process, or citations thereof. Many primary sources are provided in Kleypas, J.A., Yates, K.Y. (2009) Coral reefs and ocean acidification. Oceanography 22(4): 108-117. https://www.coris.noaa.gov/activities/oa/resources/22-4_kleypas.pdf

**Lines 196-210:**

LD: Suggest restructuring for clarity, as below.

"OA acts to alter the internal chemistry of corals and coralline algae, slowing calcification rates. The direct metabolic impacts of OA do not manifest a tipping point, but tipping points at ecological levels are likely. Recent evidence indicates that ecological tipping points within coral reefs caused solely by ocean acidification would occur around 550 ppm, roughly the same concentration of atmospheric $CO_2$ that would cause detectable declines in both coral and coralline algal calcification (Cornwall et al., 2024). However, ecosystem trajectories are uncertain, and much more future research is required to determine the generality of these findings.

The adverse impacts on coral and coralline algal calcification are direct negative effects, when combined with the direct positive effects on other taxa (such as opportunistic turfing algae). Susceptible species would start to give way to tolerant species over time (as generally occurs at natural analogues in the field (Fabricius et al., 2011, Comeau et al., 2022), and other non-coral taxa would start to dominate space on what once were traditional coral reefs. Species that are capable of maintaining stable internal carbonate chemistry or compensate for these changes tend to be more tolerant to OA.

LD: The following paragraph may be better placed at end of Interactions section.

"However, of greater immediate importance to the majority of corals will be successive marine heatwaves that will reduce the coral cover of less heat tolerant species, populations and genotypes over the majority of the oceans in the near future (van Hooidonk et al., 2014, Cornwall et al., 2021, Logan et al., 2021, Cornwall et al., 2023). Survivors of this human-driven evolutionary force will not necessarily be those that are tolerant to OA also, and thus numerous tipping points in time could occur."

**Lines 218-219 and elsewhere.**

LD: Greenhouse gas driven climate and ocean change poses a quintuple threat: ocean warming and heat waves, OA, deoxygenation, super-storms, and sea level rise - reefs having to 'catch-up' or 'drown', esp. on their lower slopes, and esp. if sea level rises rapidly and exceeds several metres over coming centuries. These are each mentioned in later sections of the manuscript.

Deoxygenation – consider mentioning that several of the 'reef gaps' in the fossil record coincide with periods of deoxygenation. Eg. See Veron, J.E.N. (2011) Mass Extinctions, Anoxic Events and Ocean Acidification. In: Hopley, D. (eds) Encyclopedia of Modern Coral Reefs. Encyclopedia of Earth Sciences Series. Springer, Dordrecht. https://doi.org/10.1007/978-90-481-2639-2_37

**Line 278:**

"Moderate rates of sea level rise may potentially aid some reefs to contend with thermal stress …"

**Line 287:**

"… , with plumes in large tropical river systems travelling many km from disturbance …"

Line 306-307 seem better placed in this Section on Pollution:

"To calculate reef change threshold exceedance, Setter et al., (2022) use an ideal value of summed proportion agricultural/urban land use <0.5 in a 50km radius around a reef."

**Lines 301-309:**

**10. Disruption**

LD: Consider a different sub-heading. The inclusion of Land use change, as a proxy for pollution, seems repetitive, as covered in section 9. Suggest focus here on Over-fishing and consider Diseases only in sub-heading 11.

**Lines 320-322:**

**11. Diseases**

"Regions such as the ==Great Barrier Reef==, the Caribbean==, the Pacific Islands, and the Indian Ocean== have been particularly impacted by these outbreaks, in some places surpassing the devastating impact of bleaching events by causing even greater coral mortality. Coral diseases stand out as being driven largely by a changing environment and are contributing ==to whole ecosystem regime shifts== (Thurber et al., (2020)...."

This statement appears too broad – perhaps 'Some areas within the GBR …'

Although diseases are becoming increasingly prevalent with temperature rise and pollution, these, by themselves, have had relatively little overall impact outside of the Caribbean Sea, to date. In the Caribbean SCTLD is a major present source of coral mortality, impacting more than a third of all reef-building coral species present, and potentially driving the extinction of Pillar coral *Dendrogyra cylindrus* (among others). The relative impact of diseases elsewhere is likely to change in future, however, becoming more prevalent, interacting with heat waves.

Two additional, relevant references.

Cavada-Blanco F, Croquer A, Vermeij M, et al. (2022) *Dendrogyra cylindrus*. The IUCN Red List of Threatened Species 2022: e.T133124A129721366. https://www.iucnredlist.org/species/133124/129721366.

Estrada-Saldívar N, Quiroga-García BA, Pérez-Cervantes E, et al. (2021) Effects of the Stony Coral Tissue Loss Disease outbreak on coral communities and the benthic composition of Cozumel reefs. Front Mar Sci 8: 632777. https://doi.org/10.3389/fmars.2021.632777

Diseases are also a major risk as 'invasive species', as more ornamental reef species are traded and transported, deliberately or accidently, across and between ocean basins. For example, several Indo-Pacific species of fish and coral have been released, mainly it seems by aquarium hobbyists, into the Atlantic, and shipping ballast water also poses significant risk as a transport mechanism. This aspect could be included in an Interactions section.

**13. Reef impact example**

**Lines 350-353:** Edit for clarity

"... large areas are becoming covered by the encrusting and bioeroding sponge Cliona spp ==(Sheppard et al., 2020 skeletons== formed a very abrasive layer on the substrate and, like liquid sandpaper, almost no larvae were seen in these areas. These sponges are clearly increasing; with one reef showing over 80% Cliona cover preventing coral larvae settlement."

This case study could be expanded to include more of the history, given it is one of the best studied remote reef systems. The paper would benefit from additional case-studies from Caribbean (nutrient enrichment, over-fishing, heat-waves, diseases, invasive species) and Great Barrier Reef (COTS, storms, run-off, heat waves, etc.), with summary noting differences in

resilience and potentially tipping points among the three systems in respect to oceanographic connectivity, coral population sizes, habitat heterogeneity etc.

**15. Resilience and adaptation**

**Lines 390-391:** Edit for clarity

"Evidence of a persistence of heat adapted genotypes at the cost of the reduction of coral diversity, i.e. the reef may survive but the biodiversity diminishes (Fox et al., (2021) Although ..."

---

## Referee Report (RR2)

**Considerations for determining warm-water coral reef tipping points.**

Paul Pearce-Kelly, Andrew H. Altier, John F. Bruno, Christopher E. Cornwall, Melanie McField, Aarón Israel Muñiz-Castillo, Juan Rocha, Renee O. Setter, Charles Sheppard, Rosa Maria Roman-Cuesta, and Chris Yesson

**Review 5ᵗʰ November 2024 Lyndon DeVantier**

LD: The authors have adequately addressed the points raised in my previous review. I have few remaining concerns, as here below.

**Abstract and 2. Considerations for assessing coral reef TPs**

Towards the end of these sections, consider including / expanding sentence(s),

a) noting that some predicted TPs have already been exceeded (albeit briefly), and that change is happening at rates faster than previously predicted, as outlined in section 14.

(b) the Abstract could also include the point that the multiple stressors are a powerful selective force – genetic bottleneck - driving significant population reductions and rapid acclimation of surviving corals (including shuffling of microbiome components) and via reproduction of survivors, the evolution of coral holobionts. The results of acclimation and adaptation will be population, species, habitat and region specific. Such evolution may, or may not, alter both the onset and rates of impact of TPs.

**5. Ocean acidification**

Line 184: Typo... material due to decreases in saturation state of CaCO3 ...

**6. Deoxygenation**

Line 230: Include sentence with more information and reference(s) to prior influence of deoxygenation in mass extinctions. Eg.

Intensified Ocean Deoxygenation During the end Devonian Mass Extinction

Jiangsi Liu, Genming Luo, Zunli Lu, Wanyi Lu, Wenkun Qie, Feifei Zhang, Xiangdong Wang, Shucheng Xie

First published: 15 December 2019

https://doi.org/10.1029/2019GC008614

Also consider noting the apparent irony that elevated SST and irradiance cause zooxanthellae to produce too much oxygen internally, causing toxicity to the coral host, while deoxygenation also linked with high SSTs causes deprivation.

**9. Pollution & disruption**

Line 308. Include sentence noting that overfishing is also linked with COTS outbreaks. See eg.

Babcock RC, Dambacher JM, Morello EB, Plagányi ÉE, Hayes KR, Sweatman HP, Pratchett MS. Assessing Different Causes of Crown-of-Thorns Starfish Outbreaks and Appropriate Responses for Management on the Great Barrier Reef. PLoS One. 2016 Dec 30;11(12):e0169048. doi: 10.1371/journal.pone.0169048. PMID: 28036360; PMCID: PMC5201292.

**12. Reef impact example**

Line 346: "... sponge Cliona spp (Sheppard et al., 2020) and ==almost no larvae were seen== in these areas."

LD: most coral larvae are not visible to the naked eye. Replace with ' ... no larvae were recorded' or 'no coral settlers were seen', depending on the original paper's findings and wording.

Line 349: '...Both sedimented surfaces and turbid water are ==hostile== to larval settlement and ==none were seen== in such areas over many hectares"

LD: Rather than 'hostile' and 'none were seen', consider 'not preferred conditions for larval settlement, with no juvenile corals recorded ...' if this better matches the original paper's findings.

**14. Resilience and adaptation**

LD: consider renaming this section as **14. Resilience, adaptation and refugia**

Line 398: Need to complete sentence: "Kleypas et al., (2021) provide a blueprint for coral reef survival and state that existing conservation measures such as marine protected areas and fisheries management are no longer sufficient to sustain reef ecosystems and many additional and innovative actions to increase reef resilience ==are needed.==

.............................

---

## Author Response (AR2)

Manuscript ESD-2023-35
**Considerations for determining warm-water coral reef tipping points.**
Paul Pearce-Kelly, Andrew H. Altier, John F. Bruno, Christopher E. Cornwall, Melanie McField, Aarón Israel Muñiz-Castillo, Juan Rocha, Renee O. Setter, Charles Sheppard, Rosa Maria Roman-Cuesta, and Chris Yesson

**Editor comments**

*While both Reviewers are positive on the merits of your submission, they also suggest a number of modifications and edits that may improve its reach and usefulness for the community. One of the Reviewers questions the subjectivity of the paper and the lack of a systematic review. I believe that this is perfectly acceptable in the context of a perspective piece. I would nonetheless ask you to provide a reasoned reply to those comments, since if your paper is published the reviews and replies will become openly accessible.*

Authors response: We thank the editor for the positive response. We have included responses to all reviewer comments below.

*As part of your revisions, try to avoid lengthening the text (or even try to shorten it if possible), as perspective pieces benefit from a concise and focused presentation. One of the Reviewers also highlights this point, and provides some specific suggestions of passages that could be cut or reduced.*

Authors response: We have revised the text and the manuscript we have cut some text where possible and added where we needed to directly address reviewer comments, the net result is the length is consistent with the previous submission.

*I am contacting you since, in making my decision on your manuscript yesterday, I forgot to attach a comment that I received from a potential reviewer, who ultimately did not accept to review your study. They based this comment purely on reading the paper abstract but I do agree that, since one typically reads the abstract to decide whether a paper is worth reading further, it is an important point to clarify. Some clarifications to this effect may also be necessary at other points in the paper, e.g. in the conclusions. I would therefore be grateful if you could include a reply to this comment too in your replies to reviewers when you submit a revised version of your manuscript.*

Authors response: We have pasted these comments and our responses below

*I see a real problem with the publication, in that we have already exceeded the thresholds that they conclude are the tipping points:*
*As of May 2022, we were already at a CO2 level of 421ppm, and in 2023, the temperature mean was 1.52C above pre-industrial levels, with 2024 running at about 1.7C. We are now in the midst of a thermal regime not expected until after 2050! See the below graph of mean ocean temperature: we clearly went over the tipping point in March of 2023. This new reality overshadows the publication and needs to be addressed.*

*The following statement is not based on our present reality:*
*"Uncertainties around tipping point sensitivities for such crucially important ecosystems underlines the imperative of robust assessment and, in the case of knowledge gaps, employing a precautionary principle favouring lower range tipping point values."*

*But perhaps the issue is that the abstract does not convey enough information?*

*More in detail: While can not be sure that we have gone over a tipping point before several years pass, in order for this paper to be relevant, it must discuss the elephant in the room: that everything is already over the predicted tipping points of the study. The study determined that the tipping point for CO2 is 350ppm, so the authors must mention that we are already at 421ppm and rapidly climbing! Assuming their study is relevant, the tipping point should already have occurred, which real-world observations confirm, despite many trying to hem and haw themselves out of accepting this new reality!*

*Based on scientific consensus, the IPCC reports clearly state that if nothing was done, that the planet would reach 1.5C above pre-industrial conditions by 2050. With the planet already exceeding 1.5C for both 2023 and 2024, we are in effect living in a thermal reality not expected until 2050. At this level of stress, the consensus is that >90% of all coral reefs would be dead. So based on what happened in 2023-24, we have already entered the great dying. To have this paper not acknowledge the present anomaly, even if not proven permanent, the paper loses most or all relevance. It gives the illusion that we are still under the thresholds if they don't clearly state what happened in 2023/24 and continues.*

[Figure]

Author response: We have added text to acknowledge that some of the tipping points that we refer to have been crossed and others will be breached in the future. We talk in the manuscript about the inherent lag in the ocean system as well as lags within reef system response times which means there is not an instantaneous cause and effect, we hope there is still time for meaningful action to avoid some of the worst case scenarios. One open question is the extent to which crossing tipping point thresholds leads to irreversible change, or whether effective measures can mitigate or reverse the tipping points. We have made edits to highlight this. We feel this issue reinforces the urgency of this message.

**Reviewer 1 - Lyndon DeVantier**

***General comments:***
*The manuscript is a very useful contribution outlining the potential tipping points arising from the many faceted issues, and interactions therein, that impact and increasingly threaten coral reefs. More information on the rationale for selection of 1.2 C and 350 ppt $CO_2$ , as per the Veron et al. (2009) paper, would be useful. The paper could benefit from an expanded Section 13 on Chagos, to include more of the history, given it is one of the best studied, remote reef systems. The paper could also benefit from additional case-studies from Caribbean (nutrient enrichment, over-fishing, heat-waves, diseases, invasive species) and Great Barrier Reef (COTS, storms, run-off, heat waves, etc.), with summary noting differences in resilience among the three systems in respect to oceanographic connectivity and variability therein, coral population sizes, habitat heterogeneity, depth ranges etc. The paper would benefit from a hard edit to shorten some sections that appear repetitive; and with shifting of some text that may fit better in other sections. The Interactions sections may be better combined into one section following all those dealing specifically with the various stressors. This would help to minimize repetition. Eg. Ocean acidification section also presently includes comments on heat waves – those could be covered solely in the Interactions section. I have suggested shuffling and shifting some text accordingly (see below Lines 196-210). Similarly in Disruptions section, some text seems to fit better in Pollution section.*

Author comments: In reference to including a separate interaction section. We had such a section in an earlier draft of the manuscript, but we chose to remove it. We felt that the separate section required a lot of repetition to reintroduce concepts. We felt it an important point to show the interactions alongside each stressor to create a more integrated whole.
In reference to the request for additional case studies. We note the editor guidelines that the manuscript in its current form requires trimming, so we don't have the space to make significant additions. We do provide a wide set of references that would allow readers to delve deeper into the Chagos and other regions if desired.

***Specific comments:***
*References could be improved by including more of the primary sources. Some examples are provided. Citations in text need to be standardized as per journal requirements.*

Author comments: We already provide an extensive reference list that includes many primary sources, but we feel the review-type publications that we cite are important to include as they are particularly relevant to tipping point considerations and act as gateways to more studies that we cannot individually cite due to space constraints.  References have been checked for standardisation.

*Lines 39-42:*
*"They are also among the most sensitive ecosystems to anthropogenic driven stressors with an estimated 50% of global live coral cover having been lost over the last 50 years (Souter et al., 2021, WWF 2022), primarily due to ocean warming (and related climate change threats of ocean acidification and deoxygenation), but in some locations also due to fishing, pollution, and disease (IPCC 2022)."*
*LD: Loss of cover – consider noting '… albeit with significant temporal fluctuations, as on the Great Barrier Reef (see AIMS LTMP reports).*
*https://www.aims.gov.au/monitoring-great-barrier-reef/gbr-condition-summary-2023-24*
*Also, Crown-of-thorns seastar (Acanthaster spp., COTS) outbreaks have caused sig. loss of coral cover across the Indo-Pacific since at least the 1960s. Heat-wave driven bleaching mortalities are increasing rapidly, since 1998. However, to date, loss of significant cover from ocean acidification*

*and deoxygenation is not well established (at least as far as I am aware), although these are serious looming threats. Nutrient enrichment (as a specific form of pollution) has had sig. impacts in the Caribbean and in parts of the Coral Triangle (eg. Areas of Java Sea).*

Author response: This section is a broad-brush introduction so there is a limit to how much detail we can provide in this section, especially when we give details on each threat later in the document. We have amended this sentence to list more examples and indicate these are a selection of many threats.

*Line 63:*
*"Approximately half the live coral cover on coral reefs has been lost since the 1870s …"*
*LD: presumably 1970s? Also recently estimated as dropping from 36 to 19 percent from 1997 to 2018 (Tebbett et al. 2023), with decreases most severe in the Western Atlantic and Central Pacific.*
*Tebbett SB, Connolly SR, Bellwood DR (2023) Benthic composition changes on coral reefs at global scales. Nat Ecol Evol 7: 71–81. https://doi.org/10.1038/s41559-022-01937-2*

Author response: We have removed the potentially confusing reference to 1870. We thank the reviewer for the additional information, but the percentage stats of 36-19% refers to hard coral cover measured at a wide selection coral reef sites, which is a different metric to the global declines we are reporting here.  We feel the important message here is the big picture global decline, so we have stuck with what we had.

*Lines 67-70:*
*"… regional die-offs (e.g. Western and Central Indian Ocean, Great Barrier Reef, Mesoamerican Reefs) (Le Nohaïc et al., 2017; Amir 2022; Muñiz-Castillo et al., 2019; Obura et al., 2022; Sheppard et al., 2020), with most reef regions having experienced multiple die-off events (Darling et al., 2019; Cramer et al., 2020; IPCC 2022). Coral reef bleaching tipping points have already been reached in seven ocean systems (IPCC 2022)."*
*LD: On GBR, Earth's largest and arguably oceanographically best-connected reef system, and elsewhere, the 'die-offs' (coral cover losses) since the 1960s (COTS mainly) have been interspersed with recovery of cover, as has again recently happened, but prior to severe 2024 mass bleaching. https://www.aims.gov.au/monitoring-great-barrier-reef/gbr-condition-summary-2023-24*
*This is not to downplay the seriousness of present impacts and future risk, nor the shifts in community structure (eg. See Richards ZT, Juszkiewicz DJ, Hoggett A (2021) Spatio-temporal persistence of scleractinian coral species at Lizard Island, Great Barrier Reef. Coral Reefs 40: 1369-1378. https://doi.org/10.1007/s00338-021-02144-4), but to add some nuance to the statements.*

Author response: We have amended this section to add nuance using the reference provided as an example of sites bucking the widespread negative trend.

*Line 100: "…is over centennial time (IPCC 2021).*

Author response: changed for clarity to "... is more than a century"

*Lines 104-105:*
*"(IPCC 2022). Overshoot of multidecadal time spans imply severe risks and irreversible impacts in many ecosystems Meyer et al. (2022), including …*

Author response: changed for clarity to "Overshoot of multiple decades implies…"

*Lines 110-111:*

*"Increasingly warmer ocean temperatures, driven by Anthropogenic climate change, compounded by El Niño heating events, is the primary stressor of regional scale mortality of scleractinian corals,.*
*LD: … and ocean-basin …*

Author response: added "… and ocean-basin…" as suggested, to show this operates at a larger scale.

*Line 113:*
*Primary sources would be better: Eg. Cite Barbara Brown and John Ogden, Peter Glynn, Ove Hoegh-Guldberg, among others, here.*
*Glynn, P.W., D'Croz, L. (1990) Experimental evidence for high temperature stress as the cause of El Niño-coincident coral mortality. Coral Reefs 8: 181–191. https://doi.org/10.1007/BF00265009*
*Brown, B.E. and Ogden, J.C. (1993) Coral Bleaching. Scientific American 268 (1): 64-70. https://doi.org/10.1038/scientificamerican0193-64*
*Glynn, P.W. (1996) Coral reef bleaching: facts, hypotheses and implications. Global Change Biology 2 (6): 495-509. https://doi.org/10.1111/j.1365-2486.1996.tb00063.x*
*Hoegh-Guldberg, O. (1999) Climate Change, Coral Bleaching and the Future of the World's Coral Reefs. Marine and Freshwater Research 50: 839-866. http://dx.doi.org/10.1071/MF99078*

Author response: We feel the most relevant references are already provided. Given the request from the editor to avoid manuscript bloat, we have not added these additional references.

*Line 143: "Since the first bleaching event of 1998,…*
*LD: Pedantic, but this should be '… first global bleaching event …' Bleaching had been documented since at least 1960s following flooding (Tom Goreau Snr).*

Author response: corrected to "Since the first underline{global} bleaching event…"

*Line 145-146:*
*"With repeated events, loss of sensitive corals and acclimation and adaptation, the DHW thresholds may change (Lenton et al., 2023)*
*LD: Also see and cite:*
*van Woesik R, Kratochwill C (2022) A global coral-bleaching database, 1980–2020. Sci Data 9: 20. https://www.nature.com/articles/s41597-022-01121-y*
*van Woesik R, Shlesinger T, Grottoli AG, et al. (2022) Coral-bleaching responses to climate change across biological scales. Glob Chang Biol 28: 4229-4250. https://doi.org/10.1111/gcb.16192*
*Donner SD, Rickbeil GJ, Heron SF (2017) A new, high-resolution global mass coral bleaching database. PLoS One 12:e0175490. https://doi.org/10.1371/journal.pone.0175490*
*LD: The time series suggested a possible increase in coral thermal tolerance – see Virgen-Urcelay and Donner 2023, Shlesinger and van Woesik 2023.*
*Shlesinger T, van Woesik R (2023) Oceanic differences in coral-bleaching responses to marine heatwaves. Sci Total Environ 871: 162113. https://doi.org/10.1016/j.scitotenv.2023.162113*
*Virgen-Urcelay A, Donner SD (2023) Increase in the extent of mass coral bleaching over the past half-century, based on an updated global database. PLoS ONE 18: e0281719. https://doi.org/10.1371/journal.pone.0281719*

Author response: We thank the reviewer for these additional references, but we do not feel they add sufficiently important details to the information we are delivering in this section, which is focussed on the number and scale of bleaching events.

*Lines 182-184:*

*The introduction to ocean acidification could benefit from a more detailed explanation of the process, or citations thereof. Many primary sources are provided in Kleypas, J.A., Yates, K.Y. (2009) Coral reefs and ocean acidification. Oceanography 22(4): 108-117. https://www.coris.noaa.gov/activities/oa/resources/22-4_kleypas.pdf*

Author response: Thank you for these suggestions, we have substantially revised this section following reviewer recommendations.

*Lines 196-210:*
*LD: Suggest restructuring for clarity, as below.*
*"OA acts to alter the internal chemistry of corals and coralline algae, slowing calcification rates. The direct metabolic impacts of OA do not manifest a tipping point, but tipping points at ecological levels are likely. Recent evidence indicates that ecological tipping points within coral reefs caused solely by ocean acidification would occur around 550 ppm, roughly the same concentration of atmospheric CO2 that would cause detectable declines in both coral and coralline algal calcification (Cornwall et al., 2024). However, ecosystem trajectories are uncertain, and much more future research is required to determine the generality of these findings.*
*The adverse impacts on coral and coralline algal calcification are direct negative effects, when combined with the direct positive effects on other taxa (such as opportunistic turfing algae). Susceptible species would start to give way to tolerant species over time (as generally occurs at natural analogues in the field (Fabricius et al., 2011, Comeau et al., 2022), and other non-coral taxa would start to dominate space on what once were traditional coral reefs. Species that are capable of maintaining stable internal carbonate chemistry or compensate for these changes tend to be more tolerant to OA.*

Author response: We have added a paragraph following these recommendations.

*LD: The following paragraph may be better placed at end of Interactions section.*
*"However, of greater immediate importance to the majority of corals will be successive marine heatwaves that will reduce the coral cover of less heat tolerant species, populations and genotypes over the majority of the oceans in the near future (van Hooidonk et al., 2014, Cornwall et al., 2021, Logan et al., 2021, Cornwall et al., 2023). Survivors of this human-driven evolutionary force will not necessarily be those that are tolerant to OA also, and thus numerous tipping points in time could occur."*

Author response: moved to the end of the interaction paragraph of the OA section as suggested.

*Lines 218-219 and elsewhere.*
*LD: Greenhouse gas driven climate and ocean change poses a quintuple threat: ocean warming and heat waves, OA, deoxygenation, super-storms, and sea level rise - reefs having to 'catch-up' or 'drown', esp. on their lower slopes, and esp. if sea level rises rapidly and exceeds several metres over coming centuries. These are each mentioned in later sections of the manuscript.*
*Deoxygenation – consider mentioning that several of the 'reef gaps' in the fossil record coincide with periods of deoxygenation. Eg. See Veron, J.E.N. (2011) Mass Extinctions, Anoxic Events and Ocean Acidification. In: Hopley, D. (eds) Encyclopedia of Modern Coral Reefs. Encyclopedia of Earth Sciences Series. Springer, Dordrecht. https://doi.org/10.1007/978-90-481-2639-2_37*

Author response: rather than debate whether there are 3 or 5 threats we have removed reference to triple threat and simplify to refer to this as one of several major threats

*Line 278: "Moderate rates of sea level rise may potentially aid some reefs to contend with thermal stress …"*

Author response: reworded to "Moderate rates of sea level rise may potentially provide cooling for some reefs contending with thermal stress… "

*Line 287:*
*"… , with plumes in large tropical river systems travelling many km from disturbance …"*

Author response: Correction made as suggested

*Line 306-307 seem better placed in this Section on Pollution:*
*"To calculate reef change threshold exceedance, Setter et al., (2022) use an ideal value of summed proportion agricultural/urban land use <0.5 in a 50km radius around a reef."*

*Lines 301-309:*
*10. Disruption*
*LD: Consider a different sub-heading. The inclusion of Land use change, as a proxy for pollution, seems repetitive, as covered in section 9. Suggest focus here on Over-fishing and consider Diseases only in sub-heading 11.*

Author response:  We agree there is substantial overlap between the pollution and disruption sections, so much so that we have decided to merge these sections together.

*Lines 320-322:*
*11. Diseases "Regions such as the Great Barrier Reef, the Caribbean, the Pacific Islands, and the Indian Ocean have been particularly impacted by these outbreaks, in some places surpassing the devastating impact of bleaching events by causing even greater coral mortality. Coral diseases stand out as being driven largely by a changing environment and are contributing to whole ecosystem regime shifts (Thurber et al., (2020).…"*
*This statement appears too broad – perhaps 'Some areas within the GBR …'*

Author response: we have edited to add this caveat

*Although diseases are becoming increasingly prevalent with temperature rise and pollution, these, by themselves, have had relatively little overall impact outside of the Caribbean Sea, to date. In the Caribbean SCTLD is a major present source of coral mortality, impacting more than a third of all reef-building coral species present, and potentially driving the extinction of Pillar coral Dendrogyra cylindrus (among others). The relative impact of diseases elsewhere is likely to change in future, however, becoming more prevalent, interacting with heat waves.*
*Two additional, relevant references.*
*Cavada-Blanco F, Croquer A, Vermeij M, et al. (2022) Dendrogyra cylindrus. The IUCN Red List of Threatened Species 2022: e.T133124A129721366.*
*https://www.iucnredlist.org/species/133124/129721366.*
*Estrada-Saldívar N, Quiroga-García BA, Pérez-Cervantes E, et al. (2021) Effects of the Stony Coral Tissue Loss Disease outbreak on coral communities and the benthic composition of Cozumel reefs. Front Mar Sci 8: 632777. https://doi.org/10.3389/fmars.2021.632777*

Author comments: We have included this suggested text.

*Diseases are also a major risk as 'invasive species', as more ornamental reef species are traded and transported, deliberately or accidently, across and between ocean basins. For example, several Indo-Pacific species of fish and coral have been released, mainly it seems by aquarium hobbyists, into the Atlantic, and shipping ballast water also poses significant risk as a transport mechanism. This aspect could be included in an Interactions section.*

Author response: We have included a line in the interaction paragraph to specify the link between invasives and disease.

*13. Reef impact example*

*Lines 350-353: Edit for clarity*
*"… large areas are becoming covered by the encrusting and bioeroding sponge Cliona spp (Sheppard et al., 2020 skeletons formed a very abrasive layer on the substrate and, like liquid sandpaper, almost no larvae were seen in these areas. These sponges are clearly increasing; with one reef showing over 80% Cliona cover preventing coral larvae settlement."*
*This case study could be expanded to include more of the history, given it is one of the best studied remote reef systems. The paper would benefit from additional case-studies from Caribbean (nutrient enrichment, over-fishing, heat-waves, diseases, invasive species) and Great Barrier Reef (COTS, storms, run-off, heat waves, etc.), with summary noting differences in resilience and potentially tipping points among the three systems in respect to oceanographic connectivity, coral population sizes, habitat heterogeneity etc.*

Author response: Unfortunately we do not have the space to expand the Chagos example, but we do provide references permitting a deeper dive into this area. We agree that there are many valuable case studies that could be incorporated here, but we are limited by the editors space constraint, but we note that other case study relevant publications are provided in some of the references cited.

*15. Resilience and adaptation*

*Lines 390-391: Edit for clarity*
*"Evidence of a persistence of heat adapted genotypes at the cost of the reduction of coral diversity, i.e. the reef may survive but the biodiversity diminishes (Fox et al., (2021) Although …"*

Author responses: edited to "There is Evidence of persistence of heat adapted genotypes but the loss of non-adapted corals leads to an overall loss of diversity"

**Reviewer 2**

*The manuscript presents a valuable review of the existing literature on the determinants and sensitivities of tipping point thresholds in warm-water coral reef ecosystems. The authors have clearly put a great deal of work into compiling and synthesizing the available research on this important topic.*

*However, the lack of a systematic review or meta-analysis approach means the study may be more prone to potential biases, which should be acknowledged and discussed in more depth. Importantly, the specific objectives of the study are not entirely clear from the current framing.*

Author response: We note that this is a perspective piece and not a systematic review. The distinction is important for the interpretation of this manuscript and an expectation of what should be included.

*Rather than exploring "where localized coral reef collapse aggregate" as stated in the introduction, the manuscript seems to focus more on documenting the known impacts of various coral reef stressors and their potential interactions. To improve clarity, the authors should clearly state the actual objectives of the work and the methodological approach used.*

*Since this is not a systematic review, it may be more appropriate to present it as an opinion piece or narrative review, with a well-developed caveats section that transparently addresses the limitations and potential biases inherent in the chosen approach.*

Author response: This is indeed a perspective piece and will be categorised as such by the journal, as agreed with the journal editor. We explicitly call this a "perspective piece" in the abstract. We have removed the line about collapse for clarity.

*Additionally, the resilience and adaptation section could benefit from a more cohesive structure and flow, moving beyond a simple literature summary to draw clearer conclusions.*

Author response: We have reviewed this section and made some minor edits to aid clarity.

*Providing additional examples beyond the Chagos case study, which focuses on cascading effects, could further strengthen the manuscript by highlighting examples of interacting stressors that may lower tipping point thresholds.*

Author response: While it would be nice to include more case studies, but we cannot do this without expanding the manuscript, please see further responses on this issue above.

*Overall, the topic is highly relevant and the authors have undertaken substantial work. By addressing the points above, the manuscript could be strengthened and provide greater value to the research community.*

Author response: We thank the reviewer for the kind words. We hope this will be relevant to both the academic & assessment community as well as policy makers.

***Additional Comments:***

*• Abstract:*

*o L26 & l33 : the is not a robust assignement, since not a systematic review*

Author response: This manuscript is calling for robust assessments, we are not claiming this is a robust assessment.

*• Introduction:*

*o L60/61 : the objectives are not in line with the content of the paper. The objectives should be rephrase and clarified overall.*

Author response: we agree this sentence is not aligned with the broader aims of the manuscript so we have removed this line.

*o L71 : I found the title of the section confusing, do you mean « Key indicators for predicting critical shifts in coral reef health »*

Author response: We have reworded this to "Considerations for assessing coral reef tipping points"

*o L98 to 100 : replece pers. Com by a published reference*

Author response: We have reworded this section and added published references as suggested.

*o L104 : typo at Meyer*

Author response: corrected

*• Resilience and adaptation*

*o L381 : see « section Reef impact example »*

Author response: removed as this is not required

*o L390 – sentence is not finished*

Author response: reworded (see reviewer comment above)

*o L394 – 396 : repetition between the two sentences*

Author response:

*o L420 : typo*

*Figure 2 : the figure does not bring any information, everything seems connected - a table might work better*

Author response: We feel this figure makes several points firstly that nearly all stressors are connected, secondly we show that nearly all links are synergistic, it also shows that these are not fixed and that the magnitude matters for the impact of the interaction. While these may be obvious to the reviewer we feel this is an important communication aid to the wider assessment community and policy makers who may not be as informed as the review.

---

## Author Response (AR3)

**Response to Reviewers – 2024-11-15**

Manuscript ESD-2023-35
Considerations for determining warm-water coral reef tipping points.
Paul Pearce-Kelly, Andrew H. Altier, John F. Bruno, Christopher E. Cornwall, Melanie McField, Aarón Israel Muñiz-Castillo, Juan Rocha, Renee O. Setter, Charles Sheppard, Rosa Maria Roman-Cuesta, and Chris Yesson

**Editor comments**

*I trust that your manuscript may be acceptable for publication in Earth System Dynamics following minor revisions according to the Reviewer's suggestions. I would also recommend adding a brief comment to the effect of the possible tipping points already having been crossed in the abstract and introductory sections. This is a natural question that many readers may have and that should be mentioned in the early part of the study, even though a more detailed discussion is deferred to the final section.*

Author response: We have added a point about exceeding tipping points to both the abstract and introductory sections.

**Reviewer comments**

*Review 5th November 2024 Lyndon DeVantier*

*LD: The authors have adequately addressed the points raised in my previous review. I have few remaining concerns, as here below.*

*Abstract and 2. Considerations for assessing coral reef TPs*

*Towards the end of these sections, consider including / expanding sentence(s),*

*a) noting that some predicted TPs have already been exceeded (albeit briefly), and that change is happening at rates faster than previously predicted, as outlined in section 14.*

Author response: We have included a point about TPs being exceeded in both the abstract and section 2, and have added a point about the rate of change in the latter.

*(b) the Abstract could also include the point that the multiple stressors are a powerful selective force – genetic bottleneck - driving significant population reductions and rapid acclimation of surviving corals (including shuffling of microbiome components) and via reproduction of survivors, the evolution of coral holobionts. The results of acclimation and adaptation will be population, species, habitat and region specific. Such evolution may, or may not, alter both the onset and rates of impact of TPs.*

Author response: We attempted to include something on this in the abstract but were unable to fit something in without spoiling the flow of the current text and so have decided to omit this. We have ensured that this important point is covered in the section on resilience. For example we state "There is evidence of the persistence of heat adapted genotypes but the loss of poorly adapted corals leads to a loss of diversity"

*5. Ocean acidification*

*Line 184: Typo… material due to decreases in saturation state of CaCO3 …*

Author response: corrected

*6. Deoxygenation*

*Line 230: Include sentence with more information and reference(s) to prior influence of deoxygenation in mass extinctions. Eg. Intensified Ocean Deoxygenation During the end Devonian Mass Extinction Jiangsi Liu, Genming Luo, Zunli Lu, Wanyi Lu, Wenkun Qie, Feifei Zhang, Xiangdong Wang, Shucheng Xie First published: 15 December 2019 https://doi.org/10.1029/2019GC008614*

Author response: Line added to section.

*Also consider noting the apparent irony that elevated SST and irradiance cause zooxanthellae to produce too much oxygen internally, causing toxicity to the coral host, while deoxygenation also linked with high SSTs causes deprivation.*

Author response: This is an interesting point, but we feel it might confuse the message on deoxygenation. In the section where we introduce bleaching we mention the breakdown in the coral/zooxanthellae relationship which is directly related the the above point:
"Heat stress, in combination with irradiance, results from small increases in seawater temperature above the summer maxima to which corals are acclimatised, destabilising the symbiosis between host corals and their symbiotic algae, commonly referred to as coral bleaching (Hughes et al., 2017; Houk et al., 2020; UNEP 2020; IPCC 2022). "

*9. Pollution & disruption*

*Line 308. Include sentence noting that overfishing is also linked with COTS outbreaks. See eg. Babcock RC, Dambacher JM, Morello EB, Plagányi ÉE, Hayes KR, Sweatman HP, Pratchett MS. Assessing Different Causes of Crown-of-Thorns Starfish Outbreaks and Appropriate Responses for Management on the Great Barrier Reef. PLoS One. 2016 Dec 30;11(12):e0169048. doi: 10.1371/journal.pone.0169048. PMID: 28036360; PMCID: PMC5201292.*

Author response: A note to this effect has been added to the interactions section.

*12. Reef impact example*

*Line 346: "… sponge Cliona spp (Sheppard et al., 2020) and almost no larvae were seen in these areas."*

*LD: most coral larvae are not visible to the naked eye. Replace with ' … no larvae were recorded' or 'no coral settlers were seen', depending on the original paper's findings and wording.*

Author response: edited to "no coral settlers were seen"

*Line 349: '…Both sedimented surfaces and turbid water are hostile to larval settlement and none were seen in such areas over many hectares"*

*LD: Rather than 'hostile' and 'none were seen', consider 'not preferred conditions for larval settlement, with no juvenile corals recorded …' if this better matches the original paper's findings.*

Author response: edited as suggested

*14. Resilience and adaptation*

*LD: consider renaming this section as 14. Resilience, adaptation and refugia*

Author response: Renamed as suggested

*Line 398: Need to complete sentence: "Kleypas et al., (2021) provide a blueprint for coral reef survival and state that existing conservation measures such as marine protected areas and fisheries management are no longer sufficient to sustain reef ecosystems and many additional and innovative actions to increase reef resilience are needed.*

Author response: Sentence edited.